# Demography and Genealogical Analysis of Massese Sheep, a Native Breed of Tuscany

**DOI:** 10.3390/ani14040582

**Published:** 2024-02-09

**Authors:** Lorella Giuliotti, Maria Novella Benvenuti, Giovanna Preziuso, Emilia Ventura, Pancrazio Fresi, Francesca Cecchi

**Affiliations:** 1Department of Veterinary Science, Università di Pisa, Viale delle Piagge 2, 56124 Pisa, Italy; novella.benvenuti@unipi.it (M.N.B.); giovanna.preziuso@unipi.it (G.P.); francesca.cecchi@unipi.it (F.C.); 2Veterinarian Free lance, Via dell’alberaccio 15, 56017 San Giuliano Terme, Italy; venturaemilia2g@gmail.com; 3Asso.Na.Pa (Associazione Nazionale della Pastorizia), Via XXIV Maggio 44, 00187 Roma, Italy; p.fresi@assonapa.it

**Keywords:** Massese sheep breed, demographic analysis, genealogical analysis, inbreeding, Tuscany

## Abstract

**Simple Summary:**

Sheep farming is crucial for preserving biodiversity, maintaining soil and water quality, and supporting local communities. Preserving genetic resources is vital for combating new diseases and tackling environmental changes. This study focuses on the Massese sheep, an indigenous breed which is primarily bred in Tuscany in extensive conditions using natural resources. The examination of their genealogy revealed that the data were incomplete, as a significant number of animals were recorded with one or two parents whose identity was unknown. This made it difficult to accurately evaluate the relationships among animals and to determine the extent of inbreeding. Breeders’ associations can play a crucial role in preserving genetic diversity; however, the completeness of the genealogical data needs improving.

**Abstract:**

This study investigates the genealogical and demographic trends of the Massese sheep breed in Tuscany from 2001 to 2021. The Herd Book kept by the Italian Sheep and Goat Breeders Association (Asso.Na.Pa) provided the data. The descriptive statistics were analyzed using JMP software. The pedigree parameters of a total of 311,056 animals (whole population—WP) were analyzed using CFC, ENDOG, and Pedigree viewer software. A total of 24,586 animals born in the period 2007–2021 represented the Reference Population (RP), and 18,554 animals the Base Population (BP). The demographic results showed an inconsistent trend of offspring registration. This study showed a short period of productivity for both ewes and rams, with means of 1.47 and 19.2 registered newborn ewes and rams, respectively. The genealogical analysis revealed incomplete data, highlighting inaccurate assessments of the relationships among the animals, and inbreeding with large differences among provinces. The average inbreeding coefficient in the WP was 1.16%, and it was 2.26% in the RP. The total number of inbreds was 2790 in the WP, with an average F_PED_ of 13.56%, and 2713 in the RP, with an average F_PED_ of 12.82%. The use of pedigree data is a key and economical approach to calculating inbreeding and relationship coefficients. It is the primary step in genetic management, playing a crucial role in the preservation of a breed. The regular updating of genealogical data is the first step to ensuring the conservation of animal genetic resources, and this study is compromised by the lack of such updates.

## 1. Introduction

Sheep originated in Southwest Asia from wild Asian mouflon, and then, migrated across Eurasia and Africa, settling down in different climatic conditions, which is why they show great variability in their resilience [1]. Local breeds acquired fundamental traits such as adaptability and frugality. Their conservation represents an important tool for tackling critical environmental events. In this regard, Massese sheep adapt to temperature changes by modifying their respiratory frequency and heart rate [2].

Animal genetic resources provide a crucial opportunity for the sustainable development of the livestock sector, whether in terms of increasing the food supply, mitigating and adapting to climate change, or promoting a broad range of ecosystem services [3].

The world’s animal genetic resources are continuing to erode because of the rapid changes affecting livestock production systems. The FAO states that the expansion of markets and economic globalization accelerate the loss of breeds and stimulate a tendency to concentrate on a few specific breeds [4]. Local breeds offer the opportunity to make use of marginal areas, preserve the environment, guarantee an income for inhabitants, and offer niche food products [5]. In fact, the preservation of native breeds is also crucial for the heritage of traditional products and practices that constitute typical products [6].

Genetics is of primary importance for both safeguarding and enhancing livestock, as it has a direct influence on health and productivity. 

The Massese sheep is one of the thirteen autochthonous Italian breeds registered in the Herd Book kept by the Italian Sheep and Goat Breeders Association (Asso.Na.Pa). Not all Massese breed farms are registered in the Herd Book, and those that are registered breed purebred animals. The Tuscany Region promotes the breeding of native Tuscan breeds through incentives for their preservation [7]. 

Data updated in June 2023 from the Italian National Data Base (BDN), related to both selected and not-selected animals, report 23,885 heads, of which 16,263 were reared in Tuscany. Massese sheep are also diffuse in other regions, especially in Emilia Romagna, Lazio, and Marche [8].

The primary use of Massese sheep is for milk, but lamb meat is also appreciated by consumers [2]. The Massese breed has an open brown colored fleece due to the presence of two coat color types (black and grey) mainly derived by combining the ASIP and MC1R mutations [9,10,11]. These genes control coat color variations in sheep, which is a trait of great importance in farm animals in terms of identification and attribution, which are valuable in genetic improvement and selective breeding programs [9,12].

The face is typically black, and both rams and ewes have horns which in males, are spiral shaped, and in females, are thinner [13].

Mature rams and ewes are around 85 and 77 cm in height, respectively. The live body weight of rams is around 90 kg, and of mature ewes, around 65 kg.

The Massese sheep’s head is light and proportionate, with a straight profile in females and a convex one in males. The eyes are lively and rather prominent, the ears are medium-sized and thin, and the muzzle is wide. The head is not woolly, and the neck is rather elongated and thin, light, and with little fleece. The chest is wide, and the back-lumbar line is long and straight. The female has a well attached breast under the belly, with little connective and adipose tissue, with an evident venous network and very fine and elastic skin [14] (Figure 1).

The puberty of Massese ewes occurs at 6–8 months and the first lamb at about 15–18 months. The good fertility of the breed and its typical reproductive pattern enables three lambings over a 2-year period [15].

The milk production of the Massese breed is 200–300 kg per lactation [16].

The meat production of the Massese breed consists of three types of lamb slaughtered in early autumn, late autumn, and late spring [17]. Lambs are often slaughtered after a suckling period of 30 days [18] and with a live weight of between 11 and 14 kg [19].

Knowledge of genetic variability is essential to preserve and improve biodiversity. The first step in any biodiversity preservation program and in genetic improvement strategies is to understand the demographic and genealogical structure. Despite the widespread use of modern molecular analysis, the genetic variability of a population is still evaluated through pedigree analysis in both livestock [20,21,22] and pets [23,24,25].

The demographic trend of Massese sheep was recently analyzed by Giuliotti et al. (2023) [26], highlighting a dramatic decline in the number of heads enrolled in the Herd Book and a lack of complete genealogical data.

The aim of this study was to examine the consistence of Massese sheep included in the Herd Book in the ten provinces in Tuscany in the last twenty years. Moreover, the genealogical variability in the breed was inspected.

## 2. Materials and Methods

### 2.1. Data Collection

This study analyzed the genealogical information of Massese sheep reared in Tuscany (Italy) from 2001 to 2021; data were provided by Asso.Na.Pa. and concerned ten provinces: Arezzo (Ar), Florence (Fi) (which included the data of the province of Prato), Grosseto (Gr), Livorno (Li), Lucca (Lu), Massa Carrara (MC), Pisa (Pi), Pistoia (Pt), and Siena (Si). The study did not consider the animals of the Asso.Na.Pa. nucleus of the Massese breed located in Asciano (Si) due to the specific management practices used.

The complete genealogical sampling represented the Whole Population (WP), comprising all founders, ancestors, and their offspring, while the Reference Population (RP) referred to 2007–2021 data. The Base Population (BP) represented animals with one or two unknown parents (absolute founders = ft).

### 2.2. Statistical Analysis

Descriptive analysis was performed with JMP software version 5.0 [27]. Longevity was calculated in the animals for which the date of elimination was available. The trend in the number of newborns during the experimental period was determined, along with the number of registered offspring/rams and the number of registered offspring/ewes.

Pedigree Viewer software v.6.3 was used for the genealogical data [28] to detect errors in the data files and to show the full pedigree structure in the WP and in the ten provinces. The other analyses were carried out with ENDOG v. 4.8 [29] and with CFC v.1.0 [30]. The following parameters were computed:-The pedigree completeness. This provides information on the completeness of the pedigree in each generation as well as the number of missing parental records. The percentage completeness of the pedigree was computed. This was also assessed by calculating the mean maximum number of generations, the mean complete number, and the number of equivalent completed generations [29]. ENDOG includes information on the completeness of each ancestor in the pedigree to the 5th parental generation. The number of discrete generation equivalents was determined because it is considered another way to describe pedigree information [31].-The number of inbred animals, the average inbreeding coefficient (F_PED_), and the average numerator relationship coefficient (ANR) in the WP and RP and per province. The data were filtered, and animals with missing information were excluded from the calculation of population statistics, specifically the average inbreeding (F_PED_) and the average numerator relationship coefficient (ANR). The software CFC [30] computes the ANR using the indirect method proposed by Colleau (2002) [32], and F_PED_ using Colleau’s modified algorithm [33]. F_PED_ is the probability that at any randomly drawn locus, a given individual has two identical alleles by descent [34], which was calculated by the tabular method described by Meuwissen and Luo (1992) [35]. The F_PED_ coefficient, number of inbred animals, and average inbreeding coefficient for each traced generation was calculated using ENDOG software. The distribution of the inbreeding level in the whole population was analyzed, and nine different class levels of inbreeding were considered: 0 < F ≤ 0.05; 0.05 < F ≤ 0.10; 0.10 < F ≤ 0.15; 0.15 < F ≤ 0.20; 0.20 < F ≤ 0.25; 0.25 < F ≤ 0.30; 0.30 < F ≤ 0.35; and 0.35 < F ≤ 0.40 [30].-The rate of inbreeding (ΔF) was considered to follow the variation in inbreeding over time, and was expressed per unit of time (generations, years) [32]. In the current study, the rate of inbreeding was calculated per generation using the classical formula ΔF = (F_t_ − F_t−1_)/(1 − F_t−1_), where F_t_ and F_t−1_ are the average inbreeding in the (th) generation. Individuals that were present in more than one generation were included in all those generations in which they were present.-The effective population size (N_e_) was calculated using ΔF as N_e_ = 1/(2ΔF). Small populations with shallow pedigrees, irrespectively of how N_e_ is computed, do not fit real populations well, thus leading to an overestimation of the real size of the population. To better characterize this, ENDOG gives three additional values of N_e_ by computing the regression coefficient (b) of the individual F_PED_ over (i) the number of full generations traced; (ii) the maximum number of generations traced; and (iii) the number of equivalent complete generations, and considering the corresponding regression coefficient as the increase in F_PED_ between two generations, and consequently, N_e_ = 1/(2b). The number of ancestors and the effective number of ancestors (f_a_) were also computed [32].

## 3. Results and Discussion

### 3.1. Demographic Data

The complete database comprised 31,156 animals (1328 males and 29,828 females) distributed throughout the whole of Tuscany.

The province with the highest number of animals enrolled in the Herd Book during the analyzed period was Lucca, followed by Pisa. On the other hand, Arezzo and Florence had the lowest number of animals (Figure 2). These last two provinces are a long way from the native area of Massese sheep and rear other breeds such as Sarda [8].

Figure 3 shows the trend of registered lambs. After an initial period of relative stability with a consistent range of between 400–600 heads, there was a notable increase in the period from 2008 to 2012, peaking at 2258 in 2012. Subsequently, there was a decline, reaching only 403 registered newborns by 2021.

This particular trend was previously observed by our research group [26] in the province of Pisa and can be explained by several factors. These include the crisis due to generational changes, the increase in predator populations [13], difficulties in adapting to health and hygiene regulations that were economically challenging, and the lack of necessary infrastructure [36]. Other contributing factors are attributed to the introduction of foreign breeds such as Lacaune [8] and changes in the legislative decree regarding animal reproduction, which may have resulted in difficulties in updating the database [26].

In this context, breeder associations could play a pivotal role in conserving, developing, and promoting local breeds. Apart from maintaining updated records in Herd Books, these associations store information regarding the characteristics of the breed and its products. Consequently, they can facilitate the marketing of specific products [37], such as “Pecorino Toscano”, a semi-hard Italian sheep cheese with a Protected Denomination of Origin (P.D.O.) [38]. To further expand Massese sheep farming, government funds [30] help breeders overcome the challenges encountered. The sheep dairy sector currently represents 3% of Italy’s agricultural sector and has the potential to increase its exports of typical dairy products, particularly with P.D.O. [39].

Figure 4 shows the registered number of offspring per ewe in the provinces. The overall results were consistent across all the provinces, with an average of registered offspring per ewe equal to 1.47 ± 0.83, and with 6195 ewes (68%) giving birth only once. Only two ewes were recorded to have eight deliveries. This could be attributed to difficulties in registering newborns, rather than indicating the short length of the ewes’ reproductive career. These findings do not align with typical sheep management practices, which generally involve four to five lactations [40].

The average longevity of the rams, calculated from 335 available data on males, was 8.5 ± 4.16 years, with large differences among provinces, ranging from 2.4 years in the province of Grosseto to 13.1 years in the province of Lucca (Table 1). The average longevity of the ewes (*n* = 12,951) was 7.9 ± 4.20 years, with values ranging from 5.0 years in the province of Florence to a maximum of 10.7 years in the province of Massa Carrara (Table 1). Generally, within a single flock, rams tend to remain in production for a shorter duration than females to prevent excessive inbreeding. Rams are thus often transferred from one farm to another, and a larger number of registered offspring per ram would therefore facilitate a more accurate evaluation of their genetic value.

The average number of lambs registered per ram was 19.2 ± 28.38, with only one instance reaching 230 offspring. Figure 5 shows the mean registered offspring per ram in the provinces of Tuscany excluding the province of Arezzo, which registered only one ram. The represented data suggest the underutilization of rams, with a potentially negative impact in terms of genetic value assessment, as noted in our previous investigation conducted in the province of Pisa [26].

### 3.2. Genealogical Data

Table 2 summarizes the main genealogical parameters calculated from 31,156 records. The BP included 18,554 animals (59.55% of the WP). The number of inbred animals was 2790. Additionally, 1486 groups of full siblings were identified, and the average number of full siblings per family was 2.15 (ranging from 2 to 6).

Out of the total number of 24,856 animals registered in the RP, both parents were known in 10,803 (43.5%), and 2713 animals were found to be inbred.

Figure 6 shows the entire genealogy of the population, divided into 11 traced generations. Figure 7 shows the genealogy of the animals categorized by province, with the highest pedigree depth observed in Massa Carrara (11 traced generations), followed by Lucca (9 traced generations) and Pisa and Livorno (7 traced generations). The province of Arezzo is not represented in the figure, as only one animal had registered parents.

The details of the pedigree quality of the Massese sheep within the WP are outlined in Figure 8, showing a similar level of pedigree completeness in both the dam and sire pathways. The completeness percentage ranged from 48.4% to 51.3% for the parental generation, from 10.1% to 17.3% for the grandparent generation, and from 2.0% to 5.3% for the third generation of great-grandparents. A comparison of these results with those reported in our previous work [26] shows higher completeness for the parental generation, but lower starting from the third generation of great-grandparents. Similar values were found for the parental generation, but were lower starting from the second generation compared to the Morada Nova breed [41], and they were lower in all generations compared to the Bharat Merino sheep [42] and the Marwari sheep [43].

The mean maximum number of generations and the mean number of complete generations were 0.76 and 0.45, respectively. There was an increase in inbreeding equal to 0.53% (N_e_ = 94.53) and 0.45% (N_e_ = 111.11) for the two categories, while the number of equivalent completed traced generations was 0.58, leading to an increase in inbreeding equal to 0.77% (N_e_ = 64.93). The average number of discrete generation equivalents in the WP was 0.83%, with significant variability among the different provinces, ranging from 0.09 for Arezzo (with a maximum of two generations) to 2.12 for Livorno. Massa Carrara and Pistoia also recorded values greater than 1 (1.78 and 1.43, respectively). These values are confirmed by the data reported in Table 3, where it is shown that the percentage of animals with both registered parents was 99.7% in Livorno, 75.8% in Massa Carrara, and 74.2% in Pistoia. The provinces with the lowest values were Grosseto (13.70%), Pisa (18.42%), and Florence (23.08%).

The RP had 19,824 ancestors, of which 4080 accounted for 50% of the genetic variability, while the effective number of ancestors (f_a_) was 2346. This finding supports the evidence of the underutilization of rams for breeding, as noted in the “Demographic Data” Section. The large number of founders suggests incomplete genealogical data, thus hindering an accurate assessment of relationships among the animals and their levels of inbreeding. Moreover, it does not provide a comprehensive understanding of the specific animals that made significant contributions to the genetic diversity of the RP.

In many foreign breeds, lower numbers of ancestors are reported as compared with our study, such as Mallick et al. (2020) [42] and Gowane et al. (2013) [44] in Gharat Merino sheep, Vatankhah et al. (2019) in Lori-Bakhtiari sheep [45], and Pedrosa et al. (2010) in Santa Ines sheep [46].

### 3.3. Inbreeding Coefficient

The average inbreeding coefficient among individuals in the WP was 1.16% and the average numerator relationship was 0.10%, while animals of the RP had an average inbreeding coefficient of 2.26% and an average numerator relationship of 0.13%. These results are difficult to compare with those from other studies on different breeds due to variations in the average inbreeding coefficients within populations, influenced by data completeness and pedigree depth. Similar values were observed in Marwari sheep [43], but higher values were reported in Bharat Merino sheep [42,44], Malpura sheep [47], and the Blue du Maine, Charmoise, and Solognote breeds [48].

The ANR coefficient was estimated to be 0.11% and 0.57% for the WP and RP, respectively; Gowane et al. (2013) [44] reported higher levels of ANR for Bharat Merino sheep, while Ghafouri-Kesbi (2010) [49] documented similar findings for Zandi sheep.

Table 3 and Table 4 present statistics related to the WP, detailing the average inbreeding, number of inbred animals, the ANR by province (Table 3), and the number of animals subdivided by range of F_PED_ within each province (Table 4). The provinces of Livorno, Massa Carrara, and Pistoia recorded the highest percentages of inbred animals. However, they exhibited lower average F_PED_ values for inbred individuals compared to provinces with limited pedigree data.

Out of 2790 inbred animals in the WP, 447 (16.02%) had an F value lower than 0.05, but 672 animals (24.09%) showed values higher than 0.20 (Table 4). The average F_PED_ in the inbreds was 13.56% (F_min_ = 0.19% and F_max_ = 42.97%), with no differences between both inbred males and females (13.05% vs. 13.62%, respectively). In the 2713 inbred animals of the RP population, the average F_PED_ was 12.82% (F_min_ = 0.78% and F_max_ = 42.97%).

The results by year (Appendix A) show that the percentage of animals with both parents known was high up to 2003 (approximately 74%), but then, dropped drastically in the period 2007–2011 (from 32% to 25%). Subsequently, registrations of parents increased again, reaching about 65–72% in the last three years, although the lambs registered in the same period were very low.

Understanding ancestry is crucial for devising improved mating strategies for genetic enhancement. Pairing sires with the lowest ANR coefficient with dams is an effective method to manage inbreeding, which, if excessive, can reduce genetic diversity, expose harmful recessive alleles, and cause inbreeding depression, which can all impact productivity, morphology, and especially reproductive fitness in all species [50,51,52,53].

Regrettably, the genealogical data for Massese sheep in Tuscany are significantly lacking, as reported for the province of Pisa [26]. To address these limitations, the only solution is to compute the inbreeding coefficient using genomic data. Various measures, such as runs of homozygosity (F_ROH_), offer valuable insights into inbreeding and can be employed in the absence of complete pedigree data [54].

Genomic data aid in studying inbreeding depression without the need for extensive parentage analysis over multiple generations [55,56,57], as these two inbreeding coefficients are related [42]. While studies demonstrate a slight correlation between genomic and pedigree-based inbreeding values in Massese sheep, other breeds with more complete pedigrees show a stronger correlation between these values [58]. Genealogical data represent the primary information source for assessing genetic variability in livestock populations [59]. However, genomic tools become fundamental when genealogical information is scarce.

## 4. Conclusions

The findings of this study reveal remarkable fluctuation in the Massese population, with a notable increase in births registered between 2007 and 2012, followed by a more evident decline from 2019 to 2021 due to various factors, including generational changes, an increase in predator populations, difficulties in adapting to health and hygiene regulations, and the introduction of foreign breeds.

The genealogical data were lacking, with numerous animals registered with at least one unknown parent. This made it challenging to accurately assess the relationships among animals and levels of inbreeding. Additionally, the use of breeding males was lower than expected, compromising an accurate assessment of the estimated breeding value.

Finally, this analysis emphasizes the importance of carefully conserving and managing animal genetic resources, especially for indigenous breeds like the Massese. Breeder associations play a crucial role in conserving and promoting these breeds; however, greater efforts are needed to improve the completeness of genealogical data and implement targeted breeding strategies to preserve genetic diversity.

In fact, when pedigrees are thorough and precise, they continue to be a useful tool. When utilized in conjunction with genomic data, they form an invaluable asset for the management of conservation genomics. Taking this into consideration, future studies should be based on molecular markers to complete and be combined with pedigree analysis.

## Figures and Tables

**Figure 1 animals-14-00582-f001:**
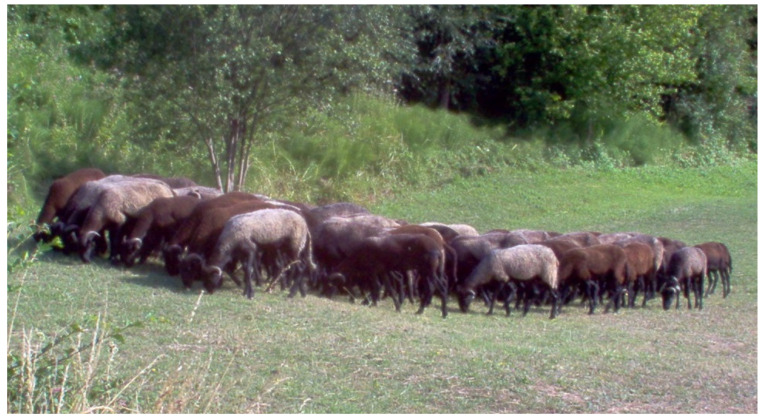
A flock of Massese sheep.

**Figure 2 animals-14-00582-f002:**
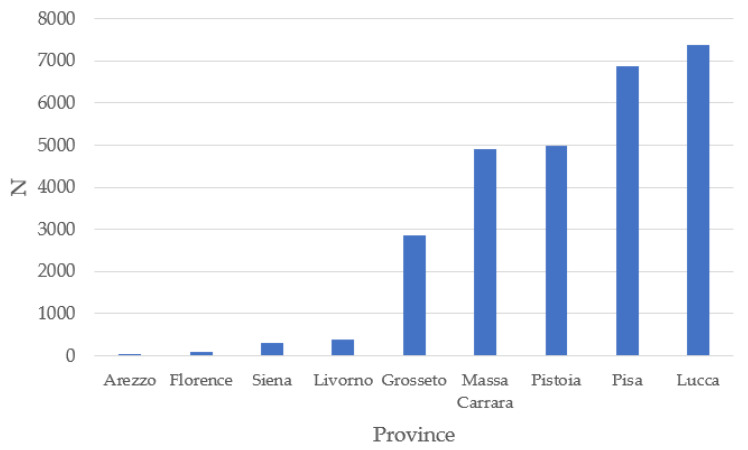
Animal distribution in Tuscany.

**Figure 3 animals-14-00582-f003:**
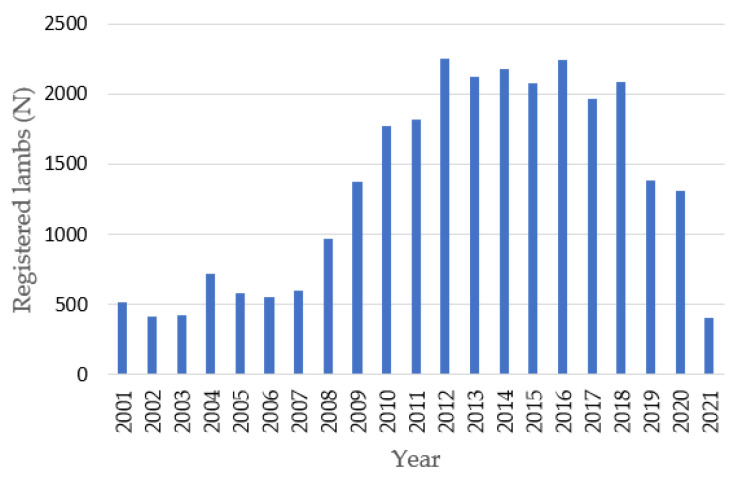
Distribution of registered lambs per year.

**Figure 4 animals-14-00582-f004:**
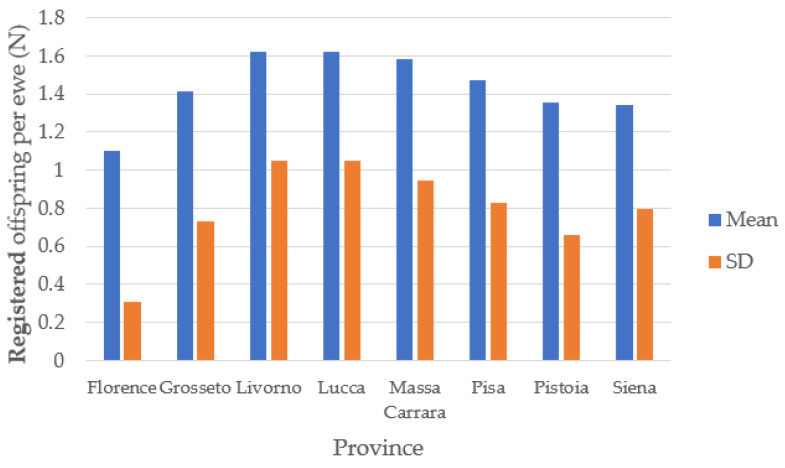
Registered offspring per ewe in Tuscany.

**Figure 5 animals-14-00582-f005:**
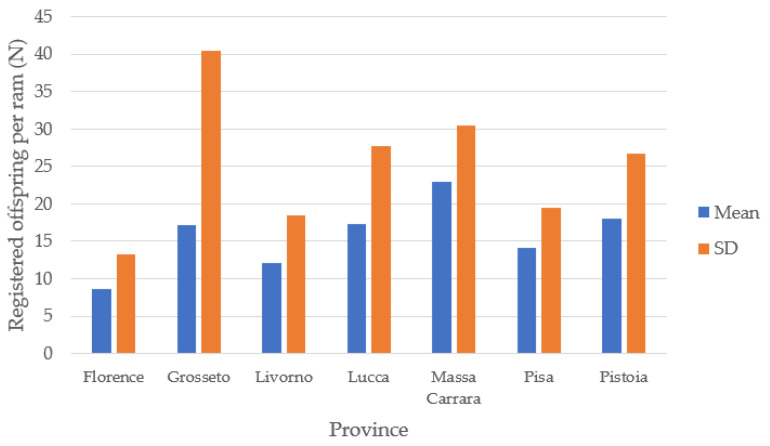
Registered offspring per ram in Tuscany.

**Figure 6 animals-14-00582-f006:**
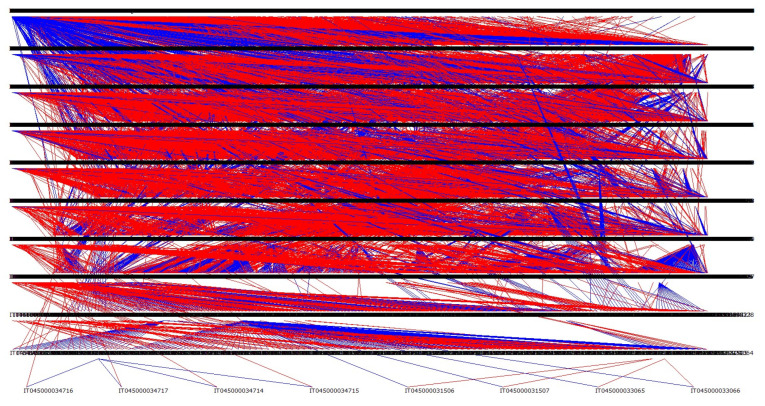
Graphic representation of the whole genealogy (sires are indicated with blue lines and dams with red lines).

**Figure 7 animals-14-00582-f007:**
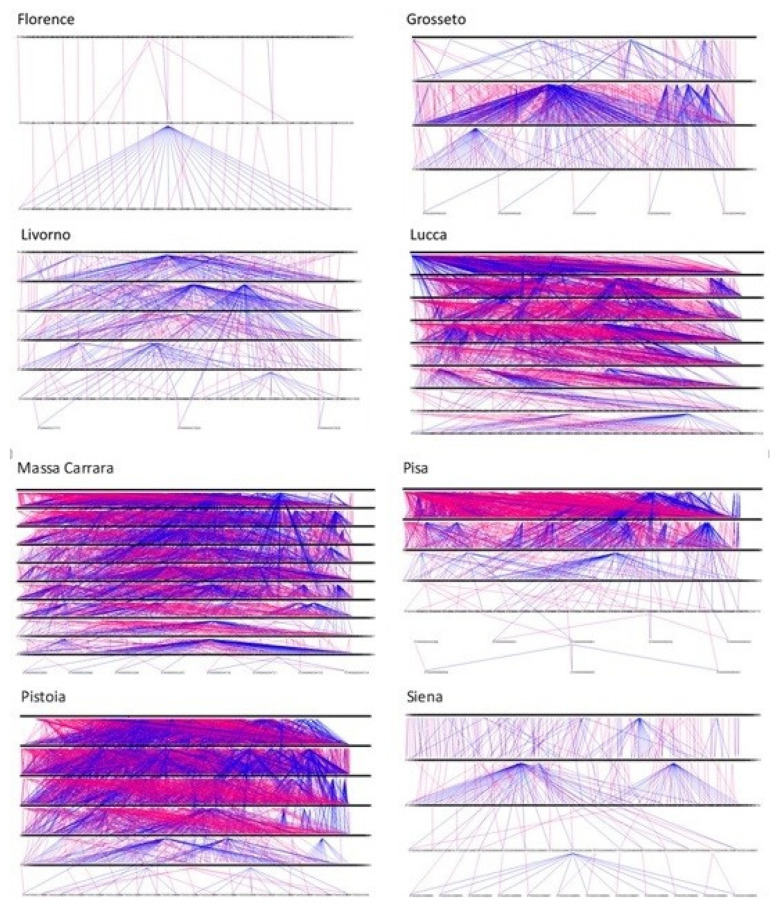
Graphic representation of genealogy (sires are indicated with blue lines and dams with red lines) for provinces.

**Figure 8 animals-14-00582-f008:**
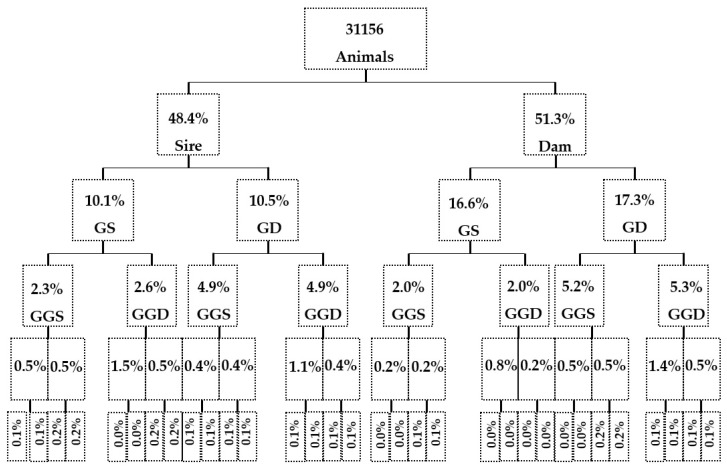
Pedigree completeness level in the whole set of pedigree data (GS and GD, grandparents; GGS and GGD, great-grandparents).

**Table 1 animals-14-00582-t001:** Longevity of parent ewes in Tuscany.

Province	Longevity					
	Rams			Ewes		
	N	Mean	SD	N	Mean	SD
Arezzo				1	7.3	-
Florence	2	6.0	2.99	42	5.0	2.47
Grosseto	42	2.4	1.63	1093	7.1	3.27
Livorno	24	5.7	4.63	216	8.3	3.39
Lucca	54	13.1	5.36	3148	8.7	4.21
Massa Carrara	81	11.5	6.90	2082	10.7	5.28
Pisa	17	12.4	6.47	3954	8.4	3.73
Pistoia	114	5.0	4.39	2344	7.1	3.13
Siena	2	5.0	0.52	71	7.0	2.37

**Table 2 animals-14-00582-t002:** Main genealogical parameters computed for the Massese sheep pedigree.

WP (n):	31,156
Males	1328
Females	29,828
Sires	660
Dams	12,662
Number of individuals with both parents known	12,602
Inbreds	2790
Full-sib groups:	1486
-Average value	2.15
-Max	6
-Min	2
BP:	18,554
-Both parents unknown	17,710
-Only mother known	784
-Only father known	60
RP 2007–2021	24,586
Number of individuals with both parents known	10,803
Inbreds	2713

**Table 3 animals-14-00582-t003:** Population statistics on average inbreeding in animals registered in each province (2001–2021).

Province	Herds (n)	Registered Animals (n)	Animals with Both Parents Known (n)	Average F (%)	ANR (%)	Inbred Animals (n)	Average F for Inbreds
Arezzo	1	23	1	0.54	0.00	1	0.125
Florence	1	104	24	0.23	1.90	1	0.240
Grosseto	12	2847	527	0.19	0.20	29	0.188
Livorno	1	376	375	3.90	11.6	127	0.115
Lucca	32	7390	2793	1.60	0.40	797	0.144
Massa Carrara	21	4908	3722	3.10	1.50	1205	0.128
Pisa	26	6879	1267	0.05	0.06	17	0.192
Pistoia	14	4983	3698	1.70	0.50	614	0.138
Siena	3	297	195	0.30	1.60	15	0.052

**Table 4 animals-14-00582-t004:** F distribution in the RP and in each province.

Range	RP	Province								
	(n°)	Ar	Fi	Gr	Li	Lu	Ms	Pi	Pt	Si
0.00 < F < 0.05	447			5	29	108	198	2	98	7
0.05 < F < 0.10	632				36	160	312	1	115	8
0.10 < F < 0.15	720	1		6	36	178	295	6	202	
0.15 < F < 0.20	319				8	106	156		49	
0.20 < F < 0.25	492		1	18	11	165	160	7	130	
0.25 < F < 0.30	95				1	39	47	1	7	
0.30 < F < 0.35	48				5	14	20		9	
0.35 < F < 0.40	34				0	25	5		4	
0.40 < F < 0.45	3				1	2				

## Data Availability

The data are contained within the article.

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
