# Peer review of "Demography and Genealogical Analysis of Massese Sheep, a Native Breed of Tuscany"

_animals, 2024, doi:10.3390/ani14040582_

Round 1

Reviewer 1 Report

Comments and Suggestions for Authors

Animals-2797299

The authors performed a genealogical analysis of indigenous Massese sheep in Tuscany for the last 20 years. The manuscript fits with the scope of the journal Animals and could be interesting to its readers. However, there are multiple issues to be fixed. I have provided some of my concerns in the following comments and many others in the annotated PDF. I believe they will help improve the manuscript. 

Title:  Demography and genetic analysis of ....... You haven't done any genetic analysis in the study. All your results are based on the genealogical analysis of the pedigree data. The title misleads the readers. Therefore, instead of the genetic analysis, it should be genealogical analysis. 

The major limitation of the study is relying on the pedigree data only and that is also not complete. Therefore, this needs to be clearly stated as the limitation of the study.  

Abstract: Genome-based pedigree analysis would have given better insight in such cases when you have incomplete pedigree data. In light of Giuliotti et al 2023, you already know (lines 85-87) that the Massese sheep lack complete genealogical data. But, still, you chose to do genealogical analysis alone. I suggest the authors acknowledge this very important limitation of the study. A sentence even in the abstract would be important. 

Line 37: Regular monitoring is essential ....... Is regular monitoring alone enough for the conservation of genetic resources? How does it support in enhancing the genetic variations? Make the sentence complete. 

Introduction:

Line 50: The world's animal genetic .......... This statement is important. Can you please provide some relevant examples of livestock? It would support your statement. 

Lines 61 -64: The information provided in the entire paragraph lacks the citations of relevant literature. 

Line 88: ... to examine the trend of Massese sheep.............. trend of what? Write specifically. 

There are numerous instances of inappropriate use of hyphens throughout the manuscript. Please check carefully. 

Results:

Line 175-176L ....... marketing of specific products such as "Pecorino Toscano" cheese....... Can you please elaborate a bit about this specific product? Non-native readers would benefit from this. 

Consider improving the visualization of Figure 7, specifically the lowermost layer.

Line 277: Results by year (data not reported).................... better include in the annex/supplement.

Lines 297-299: Despite availability of ....... This could be argued. As I mentioned already, it is worthy to acknowledge the limitations of the study. 

Conclusions: 

Please consider rewriting the entire conclusion section. There are several of unnecessary information, some vague terms, and a lack of clarity in some sentences. I have highlighted in multiple areas. 

Please find the annotated PDF attached. 

All the best!

Comments on the Quality of English Language

Minor corrections in the language are required. Inappropriate uses of hyphens were detected. 

Reviewer 2 Report

Comments and Suggestions for Authors

The study presents a very interesting aspect, the pedigree based genetic analysis of an indigenous Italian sheep breed. The results are related with the conservation and therefore the study worths publication. The statistics part is well described. Nevertheless, there are several points that have to be addressed and corrected, as follows

Although molecular genetics provide extremely valuable data regarding inbreeding and genetic diversity in a conservation point of view, pedigree analyses are also very useful. Taking this into consideration, future studies should be based on molecular markers to complete and be combined with pedigree analysis. I suggest the authors to mention this statement in their conclusion and also delete the sentence “pedigree information remains the most basic method” (line 320-321). This is also in agreement with the part in lines 289-290.

An important limitation is the lack of sufficient comparisons of the results. The authors combined Results and Discussion sections, but in several parts only results are presented. For instance section 3.2 is clearly Results. Although the authors in lines 260-263 provide a slight explanation for the absence of comparisons, this is not sufficient. Indeed, the numbers cannot be compared as numbers, but general comparisons have to be done, even with genetic diversity indices revealed by molecular markers in other sheep breeds

line 20: to accurately is duplicated, please correct

line 45-46: This sentence does not fit well here, it should transferred and grouped with other parts presenting the Massese breed.

lines 61-72: The authors describe the morphology of Massese sheep at this point, hence a figure would be welcome and indicative. Also, there is no reference for this part at all, apart from one in the end. All these presented data need at least 1-2 references more

Is the coloration of all the animals stable or is there a variability? This is important since traditionally genetic selection has been occasionally based on the colour in sheep affecting genetic diversity (https://doi.org/10.1007/s11250-022-03081-2). Please add this info and reference as well

line 93: Since there are no samplings, the term “data collection” is more appropriate 

lines 102-103: How were these animals treated? How were missing data treated? Were they excluded from the analysis?

There is also other info that should be provided and is important to be included in the manuscript, such as:

• Is the breed also reared outside Tuscany? As purebred? This is useful info, and should be added in the ms, even if not statistically evaluated

• Are all breeders of Messasa registered? And on the other hand, all registered breeders / farmers possess purebred animals?

• Is there any public authority for the maintenance of the breed apart from the breeders association? Could this be a management strategy. Please discuss this too

Round 2

Reviewer 1 Report

Comments and Suggestions for Authors

Thank you for addressing all of my comments. 

The revised titles seem more appropriate to the manuscript.

The authors have added a new picture of a flock of Massese sheep in the Introduction section. It should be labeled as Figure 1 and a caption is to be provided below the figure. Subsequent figures numbering should be adjusted accordingly. 

Line 107: . Moreover, it analysed the genealogical variability of the breed was inspected. ................. Rewrite the sentence. It has two action words. 

Line 336: The genealogical data resulted lacking -, with numerous animals registered with at...................... Complete the sentence. 

All the best!

Comments on the Quality of English Language

Minor English editing required to fix typological errors. 

Author Response

Dear Reviewer, we accepted your suggestions.  We appreciate your time and expertise. Best regards 

Lorella Giuliotti

Reviewer 2 Report

Comments and Suggestions for Authors

The authors addressed my comments, I suggest publication

Author Response

Dear Reviewer, we appreciate your time and expertise. Best regards

Lorella Giuliotti